# Ecophysiological Responses of Tall Wheatgrass Germplasm to Drought and Salinity

**DOI:** 10.3390/plants11121548

**Published:** 2022-06-10

**Authors:** Celina I. Borrajo, Adela M. Sánchez-Moreiras, Manuel J. Reigosa

**Affiliations:** 1Departamento de Bioloxía Vexetal e Ciencias do Solo, Facultade de Bioloxía, Universidade de Vigo, Campus Lagoas Marcosende s/n, 36310 Vigo, Spain; adela@uvigo.es (A.M.S.-M.); mreigosa@uvigo.es (M.J.R.); 2Agricultural Experimental Station Cuenca del Salado of INTA (National Institute of Agricultural Technology), Av. Belgrano 416, Rauch 7203, Argentina

**Keywords:** tiller density, leaf length, 13-carbon isotope, water-use efficiency, Na^+^/K^+^ ratio, proline, K^+^ concentration, Na^+^ and Cl^−^ concentration, combined stress, C3 grass

## Abstract

Tall wheatgrass (*Thinopyrum ponticum* (Podp.) Barkworth and D.R. Dewey) is an important, highly salt-tolerant C3 forage grass. The objective of this work was to learn about the ecophysiological responses of accessions from different environmental origins under drought and salinity conditions, to provide information for selecting superior germplasm under combined stress in tall wheatgrass. Four accessions (P3, P4, P5, P9) were irrigated using combinations of three salinity levels (0, 0.1, 0.3 M NaCl) and three drought levels (100%, 50%, 30% water capacity) over 90 days in a greenhouse. The control treatment showed the highest total biomass, but water-use efficiency (WUE), δ^13^C, proline, N concentration, leaf length, and tiller density were higher under moderate drought or/and salinity stress than under control conditions. In tall wheatgrass, K^+^ functions as an osmoregulator under drought, attenuated by salinity, and Na^+^ and Cl^−^ function as osmoregulators under salinity and drought, while proline is an osmoprotector under both stresses. P3 and P9, from environments with mild/moderate stress, prioritized reproductive development, with high evapotranspiration and the lowest WUE and δ^13^C values. P4 and P5, from more stressful environments, prioritized vegetative development through tillering, showing the lowest evapotranspiration, the highest δ^13^C values, and different mechanisms for limiting transpiration. The δ^13^C value, leaf biomass, tiller density, and leaf length had high broad-sense heritability (H^2^), while the Na^+^/K^+^ ratio had medium H^2^. In conclusion, the combined use of the δ^13^C value, Na^+^/K^+^ ratio, and canopy structural variables can help identify accessions that are well-adapted to drought and salinity, also considering the desirable plant characteristics. Tall wheatgrass stress tolerance could be used to expand forage production under a changing climate.

## 1. Introduction

Climate change, as well as the reduction in water reserves and the increase in salinized lands, poses major constraints to agricultural–livestock production systems [1,2]. Wild species are extremely rich resources of genetic variability for abiotic stress tolerances that are not available in cultivated species, which could help improve crop productivity and environmental sustainability [2,3]. This may be the case for tall wheatgrass (*Thinopyrum ponticum* (Podp.) Barkworth and D.R. Dewey) [3,4,5], which has been identified as a salt-tolerant forage grass [6,7,8] with potential as a soil phytoremediator [7] and energy crop [9,10]. Moreover, it has been used as a source of genes to improve tolerance to biotic and abiotic stress in wheat [3,4,5,6]. Tall wheatgrass outperforms most species in tolerance to drought [7,10] and salinity [3,4,5,6,11]. However, the physiological mechanisms responsible for the high tolerance to water and salt stress have been less extensively studied at the intraspecific level in tall wheatgrass [9,12,13,14,15].

Tall wheatgrass is mainly used as forage for direct grazing, hay, and silage in areas with salinity or alkalinity limitations in different parts of the world [6,7,9,12,16,17]. Forage grass improvement programs seek to increase its aerial biomass production and tolerance to drought and salinity. However, drought and salinity both cause an initial phase of osmotic stress, to which plants respond by minimizing water loss through stomatal closure, hence limiting transpiration, gas exchange, photosynthesis, and leaf growth [18,19]. The photosynthetic response to drought and salinity stress is very complex and is affected by both the restriction of CO_2_ diffusion into the chloroplast, through limitations on stomatal opening and the mesophyll transport of CO_2_, and the alterations to leaf photochemistry and carbon metabolism [18,20]. When stress conditions persist, salt can cause a nutritional imbalance, toxicity, and endogenous oxidative stress, severely reducing growth and development [18,19,21,22,23]. Plants activate different mechanisms to overcome stress, limiting transpiration through processes including stomatal closure and premature leaf senescence [18,19,20,21]. They also activate mechanisms to recover an osmotic balance, namely the cellular accumulation of inorganic solutes such as K^+^, Na^+^, and Cl^−^ [19,23,24] or organic solutes such as proline [14,25,26]. Proline can also act as an osmoprotective solute against oxidative stress [25,26].

Combined water and salt stress causes a severe burden and different responses among plant species [7,11,17,19,27]. The stable isotopes of carbon (^12^C, ^13^C) and nitrogen (^14^N, ^15^N) in dry matter can help us better understand the carbon and nitrogen metabolism of plants under abiotic stress [27,28]. Plants discriminate against the heavier carbon isotope (^13^C) during photosynthesis (C3-Rubisco), and this discrimination depends on the intercellular to atmospheric CO_2_ ratio in the photosynthetic organs, which varies with the stomatal conductance and intrinsic photosynthetic capacity of the C3 plant [29]. The ^13^C isotope composition (δ^13^C) provides information regarding the long-term transpiration efficiency (WUEi = A/E, photosynthesis/transpiration rate) and correlates with the WUEi and yield in C3 crops [26,29,30,31,32,33]. Furthermore, δ^13^C generally has a higher broad-sense heritability than yield and biomass [30,31,32,34]. The ^15^N isotope composition (δ^15^N) is directly related to the N metabolism, and both are negatively affected by drought and salinity, which may be due to the decreased nitrate reductase and glutamate synthetase activity [27,28]. Therefore, δ^13^C and δ^15^N values may be adequate indicators for selecting genotypes tolerant to drought or/and salinity in C3 grass [27,28,32].

In Argentina, tall wheatgrass was introduced during the 1950s for cattle grazing due to its plasticity and producibility in environments with alkalinity, salinity, summer droughts, or winter floods; currently, tall wheatgrass is used for cattle grazing on 1 million hectares of hydro-halomorphic soils in the Salado River basin [16]. Tall wheatgrass became naturalized in semi-arid to temperate humid environments three decades ago in Argentina, and variations in certain characteristics have been detected that are of agronomic interest [35,36]. Studies have been conducted to explore tolerance to drought and/or salinity at the intraspecific level in tall wheatgrass, which is especially important due to the environment in which it is cultivated. In a previous study, wild populations from contrasting environmental origins showed different levels of biomass production when subjected to different intensities of drought (100%, 50%, and 30% field capacity) over 35 days [13]. The populations presented different growth strategies (size and number of leaves and tillers) as well as differences in morpho-physiological mechanisms (leaf water content, water use efficiency, and leaf proline and protein content) to tolerate stress due to drought. This study showed that a higher biomass production stability was related to a higher tiller density and water use efficiency (WUE) [13]. In another study, populations were evaluated in terms of their salinity tolerance using two trials of different durations (40 days and 85 days) and intensities (irrigation with five salt concentrations: 0, 0.1, 0.2, 0.3, and 0.4 M NaCl), and interpopulation differences were found in biomass production according to the duration of stress [8]. In the 40-day salt trial, the interpopulation differences were similar to those found in the drought trial (differences in morphogenetic and structural characteristics, specific leaf area, and leaf proline and protein content), and differences in dead biomass were also detected. Meanwhile, in the 85-day salt trial, the interpopulation differences were attributed to differences in structural variables, leaf chlorine concentration, WUE, and ^13^C isotope level [8]. Experiments combining drought and salinity stress have not yet been carried out in this germplasm, but they are especially necessary because tall wheatgrass pastures are used for cattle grazing in environments with water and salt stress. Moreover, it is important to analyze the broad-sense heritability (H^2^) of each characteristic. The H^2^ is an estimate of the relative contributions of the differences due to genetic effects (genotypic variability) with respect to the measured phenotypic variability [30,31,32,34,37].

The aim of this study was to compare growth strategies and physiological mechanisms under drought and salinity conditions in tall wheatgrass germplasm, in order to extend our knowledge regarding tolerance to combined stress and the most effective variables for determining the germplasm that is most adapted to stress. Therefore, in this study, we examined the responses of the morpho-agronomic, physiological, and isotopic variables in four accessions of different environmental origins (P3 from a humid temperate climate and non-saline, non-alkaline soil; P4 from a semi-arid climate and saline–alkaline soil; P5 from a semi-arid climate and non-saline, non-alkaline soil; and P9 from a humid temperate climate and saline–alkaline soil) under combinations of three salinity levels and three drought levels over 90 days in a greenhouse. In addition, we estimated the broad-sense heritability of each characteristic. We tested the hypothesis that there are different physiological mechanisms and growth strategies for tolerance to drought and salinity stress in tall wheatgrass, and that the variation in these behaviors depends on the intensity of the drought and salinity and the origin of the germplasm. Consequently, this research provides a new understanding of tolerance to drought and salinity in forage species and the most effective characteristics for finding the tall wheatgrass germplasm that is most well-adapted to multiple stresses. This knowledge will help to expand tall wheatgrass culture in different environments, especially in drought and salinity conditions, which will become more typical in the near future due to climate change.

## 2. Results

Total biomass, leaf and stem biomass, the proportion of spiked tillers, plant height, and photosynthesis were reduced with increasing drought and salinity. However, the tiller density, leaf length, and water-use efficiency in the moderate drought or/and salinity treatments exceeded the control. The physiological and isotopic parameters showed different behaviors under drought and salinity, especially the δ^13^C and the Na^+^/K^+^ ratio.

### 2.1. Morpho-Agronomic Variables

The results of the three-way factorial ANOVA (four accessions, three salinity levels, three drought levels, and their interactions) for the morpho-agronomic variables are presented in the Appendix A.

#### 2.1.1. Drought and Salinity Effects

There were drought and salinity interactions (WS × SS) for most of the morpho-agronomic variables (*p* < 0.05), except for LeafB and SLA, which showed drought effects (WS) and salinity effects (SS) (Appendix A). The morpho-agronomic variables showed an absence of accession × salinity × drought interactions (*p* > 0.05). The WS × SS interactions were analyzed by comparing the WS means for each SS level (Figure 1).

The canopy structure variables (LeafT and Density) showed the highest values under moderate salt or/and drought stress (LeafT in 0 WS–0.1 SS and 50 WS–0.1 SS, and Density in 0 WS–0.1 SS and 0 WS–0.3 SS). Tiller density showed different trends when drought increased for each SS level (Figure 1a). Tiller density increased with increased drought in the salt-free level, no differences were found between drought levels for moderate SS (0.1 SS), and decreased values were found in severe SS with increased WS. However, tiller density did not differ between 50 and 70 WS in any SS level (Figure 1a). Green leaf length per tiller (LeafT) decreased when WS increased, but by a different magnitude for each SS level: -35%, −21%, and −52% between 0 and 70 WS for salt-free, moderate, and severe SS, respectively. The highest LeafT values were found in moderate SS combined with 0 or 50 WS (0 WS–0.1 SS and 50 WS–0.1 SS, Figure 1a).

Leaf biomass (LeafB) and specific leaf area (SLA) decreased under drought or salinity conditions, except for LeafB in moderate salinity. Leaf biomass decreased significantly (by −42%) when drought increased from 0 to 70 WS, though it did not differ between salt-free and moderate salinity and decreased significantly in severe salinity, with a −57% reduction compared with 0 SS (Figure 2a). Specific leaf area showed the highest values in stress-free conditions for both WS and SS treatments, whereas similar values were noted between 50 WS and 30 WS and between 0.1 SS and 0.3 SS (Figure 2a).

Dead biomass (DeadB) increased with stress, while the inverse behavior was observed for the percentage of dead biomass (Dead%). The DeadB values were highest in the control treatment, decreased with drought in the salt-free treatment, were lower in the levels with drought (50 WS and 70 WS) under moderate salinity, and did not differ between drought levels in the severe salt level (Figure 1b). By contrast, the percentage of dead biomass over total biomass (Dead%) increased when drought increased, but by a different magnitude at each SS level (Figure 1b).

In general, reproductive development and biomass production showed the highest values in the control and decreased under drought and salinity conditions, although the values were similar to the control in the moderate-salt and mild-drought treatments. Plants reduced their green stem biomass (StemB) with increasing drought and salinity, with an important drop from 0 to 70 WS under 0 or moderate SS (Figure 1c). The proportion of spiked tillers (Spike%) showed similar behavior to StemB, with a notable reduction in the spikes at higher drought levels, especially in combination with severe salinity (less than 20% spiked tillers, Figure 1d). Meanwhile, plant height (Height) decreased with increased drought for 0 and 0.1 SS, but it did not differ among the 0.3 SS levels (Figure 1c). The highest total biomass (TotalB) was produced in the control (0 WS–0 SS), and growth was limited with increasing drought and salinity, though with less difference among the 0.3 SS levels (Figure 1e).

The highest accumulated evapotranspiration (ETA) occurred in the control, but the highest water-use efficiency (WUE) was found under moderate drought–salinity conditions. The ETA decreased under drought and salinity conditions (Figure 1f). Meanwhile, WUE showed the highest values at 50 WS for salt-free and 0 and 50 WS for moderate salinity conditions, with no differences found among the 0.3 SS levels (Figure 1e).

#### 2.1.2. Accession Effects

The morpho-agronomic variables showed different behaviors between tall wheatgrass accessions (*p* < 0.05, Table 1), except for the specific leaf area (SLA) and the dead biomass (DeadB) (*p* > 0.05). The behavior of the accessions could be grouped according to their environmental origin: accessions originating from environments with greater stress, such as P4 (semi-arid climate and saline–alkaline soil) and P5 (semi-arid climate and non-saline, non-alkaline soil), and accessions originating from environments with mild to moderate stress, such as P3 (humid temperate climate and non-saline, non-alkaline soil) and P9 (humid temperate climate and saline–alkaline soil). Accessions showed no interactions between salinity or drought treatments for all morpho-agronomic variables (*p* > 0.05), except for the proportion of spiked tillers (Spike%) (ANOVAs in Appendix A).

In general, the accessions from environments with greater stress showed higher values in the canopy structural variables (Density and LeafT), which P5 showed the highest values and P3 the lowest. The differences among accessions were maintained under drought and salinity conditions. The number of tillers per plant (Density) showed the highest values in P5 and P4, and P3 showed the lowest values (Table 1); meanwhile, the leaf length per tiller (LeafT) was highest in P5 and P9 and lowest in P3 (Table 1). The accessions maintained their differences when we calculated the leaf length as a function of the tiller density, showing a quadratic trend. The treatments with different intensities of drought and moderate salinity were located at the top of the curves (with greater foliar lengths and intermediate densities, symbolized by diamonds in Figure 3). Extreme behaviors were observed in the curves of P3 and P5: P3 showed a curve displaced to the left and with less height due to its lower density and foliar length, while P5 showed a curve displaced to the right and with greater height due to its greater density and leaf length (the behavior of the density and foliar length was previously mentioned in Section 2.1.1, regarding the interaction of drought and salinity, Figure 1a).

In the partitioning of assimilates (leaf:stem or vegetative:reproductive), grouping between accessions was also observed: P4 and P5, from more stressful environments, prioritized vegetative development, while P3 and P9, from environments with mild/moderate stress, prioritized reproductive development. Canopy structural variables (Density and LeafT) determined the highest green leaf biomass (LeafB) in P5 (Table 1, Appendix A). However, P3 and P9 presented the highest values for green stem biomass (StemB), while P5 was intermediate and P4 was the lowest (Table 1). The highest plant height (Height) was reached in P3, and P5 presented the lowest (Table 1). The proportion of spiked tillers (Spike%) showed different trends between accessions for each salinity level (Appendix A, p: 0.0022). In the salt-free level, P3 showed the highest Spike% (53%); under moderate salinity, P9 was the highest (55%) and P5 the lowest (34%); and in severe salinity conditions, the spike proportion decreased notably (≤10%), and no differences were found among accessions (Table 1).

The accession P4, which came from the most stressful environment, showed the highest percentage of senescent tissue. There were no significant differences in dead biomass between accessions (Appendix A, *p*: 0.2076); however, the percentage of dead biomass was different (Dead% *p*: 0. 0482), with P4 showing the highest values and P9 the lowest (Table 1).

The combination of foliar, stem, and dead biomass determined the total biomass (TotalB) value, which was highest in P5, intermediate in P3, and lowest in P4 and P9 (Table 1). In each accession, the relationship between total biomass and its fractions (LeafB, StemB, and DeadB) was notable under drought and salinity conditions. Total and leaf biomass showed an exponential and positive relationship, maintaining their differences among accessions, though they were more attenuated with higher drought and salinity (Figure 4a). The exponential relationship showed that P3 presented the lowest and P5 the highest values of LeafB, although both accessions showed the same TotalB value. Meanwhile, total and stem biomass showed a strong positive and linear relationship in all accessions (Figure 4b). This figure shows that P3 and P9 had a higher percentage of stem for the same total biomass. These differences were smaller with increasing drought and salinity.

Accessions P4 and P5, from more stressful environments, showed lower ETA values, and P5 presented the highest WUE (Figure 4c). The ETA values were lowest in P4 and P5 and highest in P9 (Table 1). The TotalB and ETA variables showed a positive relationship, in which P5 presented the upper curve (Table 1, Appendix A). Total biomass and WUE showed an exponential and positive relationship, maintaining their differences among accessions, although their values were more attenuated with stronger drought and salinity (Figure 4c). However, in the control treatments (0 WS–0 SS), all accessions were found above the curves.

### 2.2. Physiological and Isotopic Variables

The results of the three-way factorial ANOVA (four accessions, three salinity levels, three drought levels, and their interactions) for physiological and isotopic variables are presented in the Appendix A.

#### 2.2.1. Drought and Salinity Effects

Relative water content decreased significantly with increasing salinity (−23% between 0 and 0.3 SS) and drought (−9% between 0 and 70 WS). However, RWC showed no differences between the 50 and 70 WS treatments (Figure 2b). Higher RWC values were related to a higher biomass (Appendix A). The RWC and Na^+^, Cl^−^, and δ^15^N concentrations were not significantly affected by interactions among WS, SS, and/or accession (*p* > 0.05, ANOVAs in Appendix A). Physiological and isotopic variables showed no accession × salinity × drought interactions (*p* > 0.05).

The stable isotope composition of nitrogen (δ^15^N) and Na^+^ and Cl^−^ concentrations only showed differences between salinity levels (ANOVAs in Appendix A). When salinity increased, Na^+^ and Cl^−^ increased significantly (71% in Na^+^ and 76% in Cl^−^ between 0 and 0.3 SS, Figure 2c), while δ^15^N significantly decreased (−99% between 0 and 0.3 SS, Figure 2d). This behavior was also observed when relating TotalB to δ^15^N, where a grouping by saline treatments was noted, without a unique trend (Appendix A).

The foliar nitrogen concentration increased with drought and salinity, with 145%, 71%, and 18% increases between 0 and 70 WS in the salt-free, moderate, and severe salinity treatments, respectively (Figure 5a). In general, higher values of N were observed when the biomass was lower, which occurred with the increase in drought and salinity (Appendix A). The N and K^+^ concentrations, Na^+^/K^+^ ratio, δ^13^C, and free proline content presented significant interaction only in WS × SS (*p* < 0.05, ANOVAs in Appendix A). The WS × SS interactions were analyzed by comparing the WS means for each SS level (*p* < 0.05).

The K^+^ concentration and the Na^+^/K^+^ ratio showed opposite responses to higher stress. The K^+^ increased with drought but by a different magnitude at each SS level (0 SS: 54%, 0.1 SS: 38%, and 0.3 SS: 30% between 0 and 70 WS), while for moderate and severe salinity there were no differences between 50 and 70 WS (Figure 5b). On the contrary, the Na^+^/K^+^ ratio was reduced under drought and salinity conditions. At each SS level, the highest values were found when water availability was higher (0 WS–0 SS: 1.35, 0 WS–0.1 SS: 2.19, and 0 WS–0.3 SS: 2.32, Figure 5b).

The stable isotope composition of carbon (δ^13^C) increased significantly with increased drought and salinity. Total biomass showed a negative linear regression with the ^13^C isotope (δ^13^C) as drought and salinity increased (Figure 6a). The control treatment presented the lowest δ^13^C values, and the highest δ^13^C values were reached in severe salinity combined with moderate and severe drought. Drought determined greater differences between the salt-free (9.5% between 0 SS × 0 WS and 0 SS × 70 WS) and moderate-salinity treatments (8.6% between 1 SS × 0 WS and 1 SS × 70 WS) than in the severe salinity treatment (2.4% between 3 SS × 0 WS and 3 SS × 70 WS) (Figure 5c).

Proline content showed an exponential increase with drought and salinity (5733% increase between 0 WS–0 SS and 70 WS–0.3 SS), presenting the highest values in the severe salt treatment, though the 50 and 70 WS treatments were similar for this salinity level (0.3 SS) (178–175 µmol. g^−1^ DW, Figure 5d). This variable showed a logarithmic relationship with TotalB, presenting a similar trend for all accessions (Figure 6b).

The net photosynthetic rate (A) decreased when the drought and salinity increased, by a greater magnitude over time (*t* = 3, Figure 7). A was affected by both WS × SS × t interaction (*p* = 0.0130) and RH covariate (*p* = 0.0102). The three-way factorial ANCOVA results (four accessions, three salinity levels, three drought levels, and their interactions) with covariates and repeated measures by time *(t* = 3) for the net photosynthetic rate are presented in the Appendix A. The WS × SS × t interaction was analyzed by comparing the means for each time point (*t* = 3) among WS for each SS level. Only the control (0 WS–0 SS) maintained similar values of A (23.2, 25.6, and 24.2 µmol CO_2_.m^−2^. s^−1^, for t1, t2, and t3, respectively; Figure 7). The net photosynthetic rate decreased notably for the last recorded time point (t3) under severe salinity at 50 and 70 WS (5.1 and 4.5 µmol CO_2_.m^−2^. s^−1^, respectively).

#### 2.2.2. Accession Effects

Accessions P5 and P4, from more stressful environments, showed higher values of stable isotope of carbon (δ^13^C), and P5 showed the highest relation between δ^13^C and biomass. The accession P5 presented the highest values δ^13^C, while P4 was intermediate, and P9 and P3 were the lowest (Table 1). Total biomass and δ^13^C showed a linear and negative relationship in all accessions and at all stress levels (Figure 6a), with P5 presenting the highest δ^13^C compared to other accessions with the same TotalB value. LeafB and δ^13^C showed similar behavior but greater differences among accessions (Appendix A). Only the stable isotope of carbon (δ^13^C) and Na^+^/K^+^ ratio showed differences among accessions (*p* < 0.05, Appendix A). The physiological variables showed no accession effect and no interactions between accession, salinity, or drought treatments (*p* > 0.05, ANOVA in Appendix A).

The accessions from the most stressful environments showed the highest values for the Na^+^/K^+^ ratio: P4, which came from an environment with a high alkalinity/salinity, showed the highest Na^+^/K^+^ ratio, and P5, which came from non-saline and non-alkaline soil, showed the lowest value. Additionally, accessions P3 and P9 showed a high Na^+^/K^+^ ratio, with a high level of soil salinity–alkalinity in their environment of origin. The Na^+^/K^+^ ratio varied between accessions (*p*: 0.0493), though we only noted trends in the Na^+^ and K^+^ foliar concentrations (*p* > 0.05, Appendix A). P4 had the highest Na^+^/K^+^ ratio, while P3 and P5 had the lowest (Table 1).

### 2.3. Broad-Sense Heritability

The variables δ^13^C, leaf biomass (LeafB), tiller density, and leaf length (LeafT) showed a high broad-sense heritability (H^2^: 76.9%, 71.6%, 70.8%, and 70.1%, respectively), much higher than the rest of the variables. Meanwhile, with medium heritability values, the Na^+^/K^+^ ratio (H^2^: 55.0%) stood out among the other variables, such as K^+^, proline, and those related to biomass (Heigth, TotalB, StemB, and DeadB, Figure 8). The rest of the variables showed a low heritability (H^2^ < 38.0%), (Appendix A).

## 3. Discussion

### 3.1. Drought and Salinity in Tall Wheatgrass

The intensity, duration, and rate of progression of stress conditions influence plant responses under drought and salinity [18,19]. Tall wheatgrass under higher drought and salinity showed lower photosynthesis rates (A, Figure 7) and smaller and thicker leaves (lower SLA); this translated to a lower leaf biomass, but the largest reduction was in total biomass and reproductive development (compared to StemB, Spike%, and Height). Strategies to restrict water loss (lower ETP and higher WUE) and preserve RWC were recorded through osmotic adjustment mechanisms such as the increase in proline, which maintained the Na^+^/K^+^ ratio despite the foliar increase in Na^+^ and Cl^−^. However, canopy structural variables (Density and LeafT), WUE, senescent biomass (DeadB), and foliar N and K^+^ showed particular behaviors in certain moderate-stress conditions that are explained below.

Drought and/or salinity can induce osmotic stress, leading to stomatal closure, which limits transpiration, photosynthesis, and gas exchange (CO_2_); increases the δ^13^C in dry matter; and, consequently, reduces the WUEi and yield in C3 cereals under Mediterranean conditions [26,32,38]. These relationships were also found in the present study, in which biomass production was inversely related to δ^13^C and directly related to ETP and WUE (except for the control WUE) under drought and salinity conditions (Figure 4c, Figure 6a, Appendix A); similar relationships have been reported for other forage grasses under drought [39] and salinity [28].

The control plants showed the highest photosynthesis (A) and biomass, as well as the lowest δ^13^C and the highest water loss (ETP), which led to a lower WUE (due to water-wasting behavior [40]) than the plants under mild/moderate stress (water-saving plants). Therefore, the WUE values of the control treatments were plotted separately and above of the curve (Figure 4c). This means that tall wheatgrass can behave as a water-wasting or water-saving plant [40] depending on the environmental stress level. A high leaf WUE (WUEi = A/E, photosynthesis/transpiration rate) may not always translate into a higher biomass WUE (WUE = ETP/total biomass, as recorded in this study) or yield WUE (WUE_yield_ = grain yield/ETP), since each can change depending on the variability of water availability during the crop cycle [31,32,38]. The highest WUEs in the experiment were found under moderate salinity and different drought intensities (Figure 1e); even the WUE under moderate drought and salt-free conditions (50 WS–0 SS) was higher than that of the control (0WS-0SS). This may be due to a small decrease in stomatal conductance that has protective effects against stress by allowing the plant to save water and improve its WUE [18], which is consistent with our observation of a higher δ^13^C compared to the control (Figure 5c).

The leaf length per tiller was highest in this experiment under conditions combining salinity and higher water availability (0 WS–0.1 SS, 50 WS–0.1 SS, and 0 WS–0.3 SS), surpassing the control (Figure 1a and Figure 3). Salinity may modify the duration of leaf growth [23], with a longer period spent in the blade expansion phase compared to control plants [41]. This could have caused the longest leaf length and the lowest dead biomass under moderate salinity found in tall wheatgrass in this experiment. The highest tiller density was recorded in the treatments combining drought and salt-free conditions (50 WS–0 SS and 70 WS–0 SS), surpassing the control treatment. In addition, the tiller density was higher and did not decrease with increasing drought and moderate salinity, and it was also higher with mild drought and severe salinity, showing similar values to the control plants (0 WS–0 SS) in all these treatments (0 WS–0.1 SS, 50 WS–0.1 SS, 70 WS–0.1 SS, and 0 WS–0.3 SS). It is especially noteworthy that the mild drought treatments with moderate or severe salinity achieved the highest soil salinity values (EC, Figure 1f) and the lowest restrictions in the structural and growth variables. This behavior could have occurred because the Na^+^ and Cl^−^ sequestered in the vacuole functioned as osmotic agents, improving osmotic balance and cell function [19,23] (as mentioned below).

In the severe salinity treatments, as drought increased (0 WS–0.3 SS, 50 WS–0.3 SS, and 70 WS–0.3 SS), similar WUE values and a lower total biomass were found. Tall wheatgrass controls water loss very efficiently by transpiration, which is reflected through a low ETP, a high δ^13^C, and, especially, an increase in the percentage of senescent biomass (Dead%), which was more noticeable in the treatments with severe drought and salinity. In these treatments, the highest Dead% may have been due to the excessive increase in Na^+^ and Cl^−^ in the older leaves, causing toxicity and premature senescence [19,22] (as mentioned below).

Tall wheatgrass growth strategies under drought and salinity conditions included limiting reproductive development (lower Spike% and StemB, Figure 4b) and prioritizing the allocation of photo-assimilates to leaf and tiller growth (higher Density, LeafT, and LeafB) (Figure 1). Tall wheatgrass pasture management is based on maintaining the vegetative canopy by generating leaves and tillers to achieve good quality, production, and persistence; therefore, it is essential to avoid reproductive development that produces lignified plants and persistent inflorescences, which negatively affect cattle grazing [16,36]. Therefore, tall wheatgrass growth strategies under drought and salinity would facilitate the management of tall wheatgrass pastures.

Tall wheatgrass continued to grow even under severe stress due to various mechanisms of osmotic adjustment and nutrient balance that helped maintain cell functions. Proline has been mentioned as an organic solute whose accumulation is aimed at osmoregulation and osmoprotection in tall wheatgrass under both drought [13,25] and salt stress [8,14]. In the present study, the proline concentration increased together with a reduction in total biomass and an increase in drought, but the increase was exponential when drought and salinity were combined (Figure 6b). This increase has never been cited in other species, so it is probably evidence of the superior capacity of tall wheatgrass for osmotic adjustment mechanisms and protection against oxidative stress [26].

K^+^ has a prominent role in the photosynthetic process, osmotic adjustment, and stomatal movement [18,24]. The cellular K^+^ concentration and maintenance of a low Na^+^/K^+^ ratio are essential for plant growth and tolerance to drought and salinity [24]. An increase in the K^+^ concentration may be the main inorganic osmoregulation mechanism used by tall wheatgrass under drought, as it was registered in the present experiment, though the increase in K^+^ was lower under drought and salinity (Figure 5b). In other experiments with tall wheatgrass, the leaf K^+^ concentration also increased under drought [11] but decreased with mild to moderate salinity [8,11,14,15].

In tall wheatgrass, salt tolerance is associated with the restricted accumulation of Na^+^ and Cl^−^ and the maintenance of a low Na^+^/K^+^ ratio in shoots [14,15]. However, the Na^+^ and Cl^−^ sequestered in the vacuole could function as low-energy osmotic agents [19,23]. We assume that this is the main inorganic osmoregulation mechanism used by tall wheatgrass when under a combination of salinity and drought conditions. The Na^+^/K^+^ ratio decreased under salinity and drought conditions (Figure 5b), restricting growth and development (with lower photosynthesis, LeafB, StemB, and TotalB and higher δ^13^C, Dead%, and Spike%) despite the high salinity of the substrate in the moderate- and severe-salinity treatments (>9.0 dS.m^−1^ EC, Figure 1f). Nevertheless, if stress persists over time, and excessive amounts of salt enter the leaf, the vacuole is filled, and the Na^+^ or Cl^−^ content in the oldest leaves increases to toxic levels, causing premature senescence [19,22], reducing transpiration and photosynthesis, and consequently limiting growth and flowering [8,19]. This may happen in tall wheatgrass under the combined conditions of severe salinity and moderate/severe drought.

Although it is known that a water deficit substantially limits N uptake and assimilation [27,28,42], the behavior of δ^15^N and the increase in foliar N and nitrogenous compounds such as proline in our experiment indicate that nitrogen was not a limiting factor in the growth of the tall wheatgrass under different combinations of drought and salinity, even under conditions of strong stress (Appendix A). This may be in part because the plant can recycle large amounts of previously assimilated N from senescent leaves [22,42]. Previous studies in tall wheatgrass reported similar responses under salinity [8], while under drought, an increase in proteins was recorded [13,25]. This was contrary to the reports for annual grasses regarding the limitation in N metabolism with a lower foliar concentration due to the lower nitrate reductase (NR) and glutamate synthetase (GS) activity under drought and salinity [27,28]. However, the enzymatic activity of NR and GS may vary according to the species and its tolerance to salinity [43].

### 3.2. Accession Variability and Heritability

Phenotypic differences in total biomass were the product of differences in leaf, stem, and dead biomass; water-use efficiency; and physiological mechanisms such as the δ^13^C and the Na^+^/K^+^ ratio (Figure 4 and Figure 6a). These may be due to different intraspecific strategies for the allocation of photo-assimilates and different morpho-physiological mechanisms for the absorption of water and nutrients [18,19,21,22], which are discussed below.

Forage grass improvement programs seek to increase aerial biomass production and tolerance to drought and salinity. In the present study, accession P5 showed the highest aerial biomass production (TotalB), and accession P3 was intermediate in all drought and salinity treatments, though the photosynthetic rate and RWC were similar among all accessions (P3, P4, P5, P9). This was probably because they were measured in the youngest leaves, whose continued growth is always prioritized by the plant [18,19,22]. There were also no differences among accessions in leaf proline, despite the exponential increase in proline concentration between control and maximum stress treatments (Figure 6b). Our results are consistent with other studies under salt stress, which found that differences in proline were not sufficient to determine an intraspecific selection for salinity tolerance in tall wheatgrass [8,14], contrary to reports for other species [26].

Accession P5 originated from an environment with intense drought but non-saline conditions (Table 2), and its strategy was to prioritize the allocation of photo-assimilates to leaf and tiller growth (the highest Density, LeafT, and LeafB) and limit reproductive development (lower Spike% and Height, intermediate StemB) compared to the other accessions (Figure 4a,b). In addition, P5 showed the highest biomass and WUE (Figure 4c), the lowest ETA (Appendix A), and the highest δ^13^C signature (Figure 6a and Appendix A). This may have been due to an increase in leaf WUE caused by partial stomatal closure [18,38], and/or an increase in mesophyll conductance to CO_2_ [20]. Unfortunately, neither stomatal nor mesophyll conductance were recorded in this work. In wheat, landraces exhibited higher δ^13^C values than modern cultivars, which might have been a consequence of a lower stomatal conductance; these differences in δ^13^C were recorded in grains [32] and in weight dry matter per plant at anthesis and biomass at maturity [33]. This behavior is similar to that found in P5, for which a higher δ^13^C indicates a greater tolerance to stress [30,33].

Meanwhile, the accession P4, which originated from the most stressful environment, with saline–alkaline soil (Table 2), showed the lowest leaf biomass production but presented similar behavior in terms of structural variables (Density and LeafT) when compared to P5. The behavior presented by P4 may have been the result of a salinity and drought tolerance strategy that caused the accumulation of toxic ions and early senescence in old leaves (the highest Na^+^/K^+^ ratio and Dead%), reducing transpiration (the lowest ETA) and maintaining the water status and photosynthesis in younger leaves but reducing the radiation interception capability and growth of the plant (high Density with lower ETP, LeafB, StemB, and TotalB).

On the contrary, accessions P3 and P9, which originated from environments with mild or moderate stress (Table 2), showed higher reproductive development (the highest StemB, intermediate Height, and high %Spike), together with a high ETP and the lowest WUE and δ^13^C values. Therefore, accessions P3 and P9 likely prioritized reproductive development and seed production as a growth strategy for their self-perpetuation. In contrast, accessions P5 and P4, which originated from more stressful environments, likely prioritized vegetative development through increased tillering and a lower proportion of spiked tillers as a growth strategy to maintain perenniality, though with different mechanisms for controlling transpiration, leaf senescence, and the Na^+^/K^+^ ratio.

Salt tolerance in tall wheatgrass is associated with the maintenance of low Na^+^/K^+^ ratios in the shoots of tolerant lines compared to the shoots of sensitive lines [14,15]. In the present study, the accession P5, which originated from a non-saline and non-alkaline environment, showed the lowest values for the Na^+^/K^+^ ratio and the highest in total biomass, while accession P4, which originated from an environment with saline–alkaline soil, showed the contrary behavior. These findings were opposite to what was expected considering the environmental origin of the accessions, but it was important to differentiate the accessions with higher biomass production. In addition, the Na^+^/K^+^ ratio presented medium heritability (H^2^ 55.0%) and could be a useful characteristic to take into account for the selection of accessions with greater tolerance to stress.

In all drought/salinity combinations tested, tall wheatgrass showed that the highest values of δ^13^C may be adequate indicators for the selection of accessions with a greater tolerance to drought and salinity, as has been found in other grasses [30,31,33]. In the present work, the regressions between δ^13^C and total biomass (Figure 6a) or leaf biomass (Appendix A) clearly showed the differences among accessions. Therefore, it was important to estimate the broad-sense heritability (H^2^) for each characteristic to analyze what proportion of the measured phenotypic variation was due to genetic effects rather than environmental effects or genotype x environment interaction [30,31,32,34,37]. A high H^2^ indicates the high repeatability of the characteristic and the greater reproducibility of the genetic differences, as has been cited for δ^13^C in the literature [34]. In the present study, δ^13^C presented the highest heritability compared to the other characteristics, similar to what has been found in other grasses [30,31,32,34]. Furthermore, the δ^13^C value is easier to measure than other physiological variables [27,31]. The heritability of leaf biomass, density, and leaf length were also high (H^2^ > 70.0%), which is significant, as the canopy structure in forage species gains relevance according to the purpose of pasture use.

Finally, the absence of an interaction between drought, salinity, and accessions for most of the variables; the morpho-agronomic, physiological, and biochemical responses among accessions consistent with previous studies [8,13]; and the broad-sense heritability of each characteristic allow us to conclude that the combined use of the δ^13^C signature, the Na^+^/K^+^ ratio, and the canopy structural variables (leaf length and tiller density) are useful for identifying accessions which are better adapted to drought and salinity, also considering, the desirable characteristics of grass. In this study, the accessions showed different strategies and physiological mechanisms for coping with drought and salinity, which should continue to be investigated, taking into account their use in the field. Accession P5 should continue to be evaluated with a view to its use as cattle grazing forage, as it generated shorter plants with lots of tillers and the highest biomass production, prioritizing vegetative development. Meanwhile, accession P3 should be harvested for hay or biofuel production, as it generated taller plants and achieved higher biomass production, prioritizing stem development. On the other hand, for genetic improvement regarding resistance to biotic and abiotic stresses in wheat, wheatgrass germplasm has been implemented through intraspecific crosses, chromosomal segments, or genes/QTL with successful results [3,4,5]. Nevertheless, to improve tolerance to drought and salinity in wheat with the genetic basis of the tall wheatgrass germplasm evaluated herein, we should integrate physiological and biochemical mechanisms through proteomic studies, as well as the detection of the genes/alleles/QTL associated with these responses [1], which are still being studied [3,4,5,6] and transcend the present work.

## 4. Materials and Methods

### 4.1. Origin of the Germplasm and Experimental Conditions

Accessions of naturalized populations of tall wheatgrass (*Thinopyrum ponticum* (Podp.) Barkworth and D.R.Dewey) (2n = 10x = 70) were provided by the Active Germplasm Bank of the National Institute of Agricultural Technology in Balcarce, Argentina. Four accessions (P3, P4, P5, P9) were selected from contrasting climate/edaphic environments: P3 from a humid temperate climate with non-saline and non-alkaline soil, P4 from a semi-arid climate with saline–alkaline soil, P5 from a semi-arid climate with non-saline and non-alkaline soil, and P9 from a humid temperate climate with saline–alkaline soil (Table 2). Seeds were germinated in chambers (30°/20 °C and 8/16 h light/darkness), and seedlings were transplanted into small pots with a substrate (Compo Sana Universal R) in a greenhouse. After 45 days, plants with three tillers were selected and transplanted into test pots (17.7 ± 2.4 °C.day^−1^ and 12 h 35 min daylight).

The experiment was carried out in pots in a glasshouse at the Lagoas-Marcosende Campus of the Universidade de Vigo, Spain (42°10′0.38′’ N, 8°41′3.37′′ W). The experiment was took place over 90 spring/summer days, with 14 h 50 min (±25 min) of natural daylight. During the experiment, the daily temperatures in the greenhouse were recorded, reaching a mean temperature of 20.9 °C, a maximum temperature of 28.0 °C, and a minimum temperature of 13.7 °C (Appendix A). The experimental unit was the pot containing one plant with three tillers (1 plant.pot^−1^). Plants began the experiment in the vegetative stage and ended in the reproductive stage, with some of their tillers in the phase of stem elongation. A commercial mixture was used as substrate (Compo Sana Universal R; nutrient composition in mg. L^−1^: N = 325, P_2_O_5_ = 350, K_2_O = 425; pH = 5.7; salt concentration in g. L^−1^: <2.5).

Salinity treatments comprised irrigating the pots weekly with three salt treatments consisting of 0, 0.1, and 0.3 M NaCl solutions dissolved in water (EC: 0.9, 8.1, and 20.9 dS.m^−1^, respectively), simulating salt-free, moderate, and severe salt stress levels (0 SS, 0.1 SS, or 0.3 SS, respectively). Soil salinity was gradually increased through irrigation without causing osmotic shock in the plants, which was important to avoid cell plasmolysis [19]. Drought treatments were established with three levels of irrigation at 100%, 50%, or 30% water retention capacity of the pot, simulating mild, moderate, or severe drought stress level (0 WS, 50 WS, or 70 WS, respectively). Treatments were maintained by weighting the pots weekly and adding the amount of water or salt solution lost by evapotranspiration [13]. The experiment was carried out as a randomized complete block design with a 3 WS × 3 SS × 4 accession factorial arrangement (three drought levels, three salinity levels, four accessions) and five repetitions (N = 180). In total, four accessions of different origins (P3, P4, P5, P9) were subjected to nine combined treatments of drought and salinity.

### 4.2. Morpho-Agronomic Variables

Evapotranspiration (ET) was estimated as the water loss in each pot weekly. ET was calculated as the difference between the pot weight immediately after irrigation and its weight 7 days later. This was repeated over 12 weeks. ET was measured in g_H_2_O_ pot^−1^ and expressed in mL_H_2_O_ pot^−1^. Accumulated evapotranspiration (ETA, mL_H_2_O_ pot^−1^) was calculated as the sum of ET (*n* = 12) in each pot [13].

Plant height was recorded in three tillers per plant from the soil base to the highest point (Heigth, cm. plant^−1^) on day 87. Total aerial biomass (TotalB) of plants in each pot was measured on day 90 and divided into green and senescent biomass of leaf blades, pseudostem, and stems. Senescent biomass was considered dead biomass (DeadB). Green leaf blades were considered foliar biomass (LeafB), and green pseudostem and stem as stem biomass (StemB). For each biomass, dry weight (DW) was estimated after drying the forage at 50 °C until reaching a constant weight and was expressed in g DW. plant^−1^. Additionally, the percentage of dead biomass over total biomass (Dead% = DeadB/TotalB × 100) was calculated. Water-use efficiency (WUE) was calculated using the equation: WUE = TotalB/ETA, mg mL^−1^_H_2_O_ [13]. Specific leaf area (SLA, cm^2^.g^−1^), as area/dry mass of leaf blade area, was estimated using the youngest fully expanded leaf of the main tiller of each plant [44]. Leaf blade area was calculated with ImageJ software [45]. We also measured total green leaf length per tiller (LeafT, cm. tiller^−1^) in three tillers per plant and total number of spikes and tillers per plant (Spike, spike. plant^−1^ and Density, tiller.plant^−1^, respectively) [8]. Then, the proportion of spiked tillers was calculated as Spike% = (Spike/Density) × 100.

### 4.3. Physiological and Isotopic Variables

The net photosynthetic rate (A, µmol CO_2_.m^−2^.s^−1^), leaf and air temperature, air flows, CO_2_ concentration, and relative humidity (PAR, Temp_A, Temp_L, PAR, Flow, CO_2_, RH, respectively) were measured at 10, 45, and 85 days (*t* = 3), using a LI-6200 portable infrared gas analyzer (Li-cor, Lincoln, NE, USA) [44]. Measurements were registered between 11:00 am and 3:00 pm using the youngest fully expanded leaf of each plant, in four repetitions (144 records each day, 432 total observations). Atmospheric CO_2_ concentration ranged from 380 to 400 μmol.mol^−1^, and air relative humidity between 30% and 40% during data collection in the IRGA measurement chamber. Relative water content (RWC, %) was measured using the youngest fully expanded leaf of each plant per pot [44]. Leaf biomass samples were analyzed by the Bates method to determine free proline (proline, µmol.g^−1^ DW) [44]. Leaf biomass samples were used for total C and N concentration determination (*w*/*w*, dry basis%) with an elemental CHN analyzer (EA 1110 Automatic Elemental Analyzer, Fisons Instruments). Additionally, Cl^−^, K^+^, and Na^+^ contents were analyzed in a Perkin Elmer Optima 4300DV inductively coupled plasma-optical emission spectrometer (ICP-OES), and the Na^+^/K^+^ ratio was calculated. Furthermore, the stable isotope compositions of carbon ^13^C/^12^C (δ^13^C, ‰) and nitrogen ^15^N/^14^N (δ^15^N, ‰) were analyzed using automated elemental analysis coupled to an Isotope Ratio Mass Spectrometer (Thermo Finnigan MAT252, Thermo Fisher Scientific, Bremen, Germany).

### 4.4. Soil Electrical Conductivity

Finally, the pot soil was analyzed to measure electrical conductivity (EC, dS.m^−1^). In the soil, the electrical conductivity (EC) revealed the severity of the saline conditions increased by drought that the tall wheatgrass endured throughout the nine treatments (Figure 1f). All drought levels without added salt presented similar EC values at the beginning and end of the experiment (0 WS–0 SS, 50 WS–0 SS, and 70 WS–0 SS), and they were classified as non-saline substrates (<1 dS.m^−1^). Meanwhile, the EC increased notably with increasing saline irrigation, exceeding the value of 4 dS.m^−1^, which classifies a soil as high-salinity soil [17], in all levels of drought with moderate and severe salinity, with extreme values between 9.2 and 25 dS.m^−1^ in 70 WS–0.1 SS and 0 WS–0.3 SS, respectively. This value was three times higher than that reported as the highest for tall wheatgrass in the tolerance threshold to EC [7,12], which is very important considering that tall wheatgrass is a C3 perennial forage grass used in hydro-halomorphic environments.

### 4.5. Statistical Analyses

All variables were analyzed using three-way factorial ANOVA (four accessions, three salinity levels, three drought levels, and their interactions) and five blocks (180 total observations), except for net photosynthetic rate (A), which was analyzed with three-way factorial ANCOVA (PAR, Temp_A, Temp_L, Flow, CO_2_ and RH covariates) with repeated measures by time (*t* = 3) and four blocks (432 total observations). Accession, salinity, and drought were fixed effects, whereas block was considered a random effect. Means were compared using least significant difference test (LSD, *p* < 0.05). Statistical significance was defined at the 95% confidence level. Analysis was conducted using PROC MIXED/LSMEANS [46]. Proline, Dead%, Spike%, EC, and ETA were logarithmically transformed before analysis. Regressions were estimated in pairs of variables for each accession or accession mean using PROC REG and PROC RSREG [46], with the intention of showing the behavior of total biomass as a function of different variables.

The broad-sense heritability (H^2^) of all variables was determined by the estimation of variance components using Proc Mixed ratio covtest in SAS (REML) [46], considering four accessions, nine environments (resulting from the combinations of drought and salinity stress), and their interactions with five repetitions (or four, in the case of A). The H^2^ was not estimated for the ETP and WUE variables, since they were registered on the soil-plant system (being the fraction of water evaporation exclusively from the soil). The broad-sense heritability (H^2^) and its standard errors (SE H^2^) were estimated using variance ratios considering the following equations [37] on a population mean basis: H^2^ = {σ^2^ g/(σ^2^ g + σ^2^ ge/e + σ^2^/re)} × 100 and SE H^2^ = SEσ^2^ g/(σ^2^ g + σ^2^ ge/e + σ^2^/re), in which σ^2^ g = genotypic variance (accessions), σ^2^ ge = variance due to genotype by environment interaction, σ^2^ = residual variance, σ^2^ p = phenotypic variance, r = number of replicates, and e = number of environments. Heritability estimates were categorized as low (0–40%), medium (40–59%), high (60–79%), and very high (>80%) [47].

## 5. Conclusions

Our results show that tall wheatgrass prioritizes vegetative growth instead of reproductive development by increasing tiller density and leaf length as a plant strategy to remain perennial and increase survival in environments with strong drought and salinity. This strategy is beneficial to pasture management, since tall wheatgrass achieves higher quality, production, and persistence by maintaining a greater vegetative canopy.

The highest values for canopy structural variables (leaf length and tiller density) and WUE under moderate salinity–drought were attained by active inorganic osmoregulation mechanisms involving an increase in the K^+^, Na^+^, and Cl^−^ content in the vacuoles but a decrease in the Na^+^/K^+^ ratio. Additionally, we observed organic osmoregulation through an increase in proline, which functions as an important osmoregulatory and osmoprotective solute in both drought and salinity conditions. This is essential considering that wheatgrass is a C3 forage grass used in hydro-halomorphic environments.

Evidence of interactions between accessions and stress was only found in certain variables. Phenotypic variability among accessions was due to differential growth strategies and physiological mechanisms under drought and salinity conditions. The P3 and P9 accessions, belonging to environments with mild/moderate stress, prioritized reproductive development, with higher evapotranspiration and the lowest WUE and δ^13^C signature. Meanwhile, accessions P4 and P5, belonging to more stressful environments, prioritized vegetative development through tillering, with the lowest evapotranspiration; the highest δ^13^C signature; and different mechanisms to limit transpiration, leaf senescence, and the Na^+^/K^+^ ratio. Finally, considering the differences among accessions and the H^2^ of the characteristics, we concluded that the combined use of the δ^13^C value, the Na^+^/K^+^ ratio, and the canopy structural variables can help to identify accessions that are well-adapted to drought and salinity conditions also considering the desirable characteristics of grass. Tall wheatgrass germplasm could be used to expand the production of forages, biofuels, and crops in the face of climate change.

## Figures and Tables

**Figure 1 plants-11-01548-f001:**
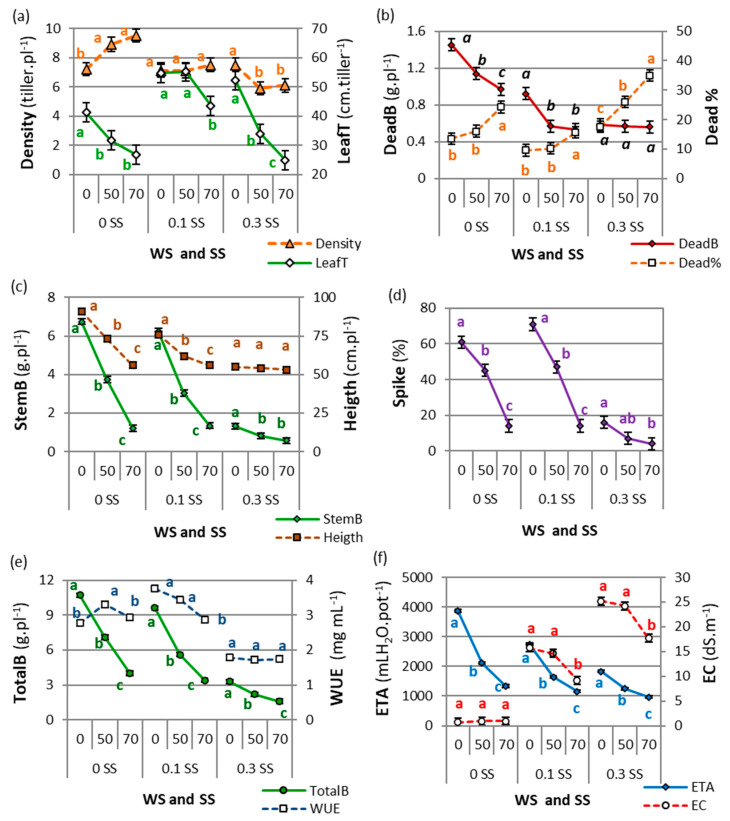
Drought and salinity interactions (WS and SS) for: tiller density and leaf length per tiller (LeafT) (**a**), dead biomass (DeadB) and percentage of dead biomass (Dead%) (**b**), stem biomass (StemB) and plant height (Height) (**c**), proportion of spiked tillers (Spike%) (**d**), total biomass (TotalB) and water-use efficiency (WUE) (**e**), and accumulated evapotranspiration (ETA) and electrical conductivity (EC) (**f**). For each variable, different letters indicate significant differences (LSD, *p* < 0.05) between drought levels (WS: 0, 50, or 70, corresponding to 100%, 50%, or 30% water capacity, respectively) for each level of salinity (SS: 0, 0.1, or 0.3, corresponding to 0.0, 0.1, or 0.3 M NaCl, respectively). Points are means of 4 tall wheatgrass accessions and 5 blocks (*n* = 20). Bars indicate the means standard error.

**Figure 2 plants-11-01548-f002:**
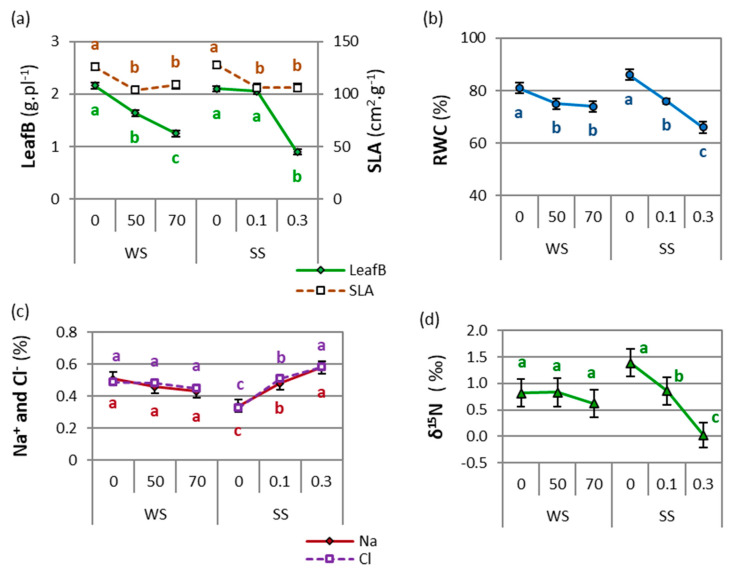
Effect of drought level (WS) or salinity level (SS) on leaf biomass (LeafB) and specific leaf area (SLA) (**a**); relative water content (RWC) (**b**); Na^+^ and Cl^−^ concentrations (**c**); and stable isotope of N (δ^15^N) (**d**). For each variable, different letters indicate significant differences (LSD, *p* < 0.05) between drought levels (WS: 0, 50, or 70, corresponding to 100%, 50%, or 30% water capacity, respectively) or salinity levels (SS: 0, 0.1, or 0.3, corresponding to 0.0, 0.1, or 0.3 M NaCl, respectively). Points are means of 4 tall wheatgrass accessions, 5 blocks, and three levels of salinity or drought (*n* = 60). Bars indicate the means standard error.

**Figure 3 plants-11-01548-f003:**
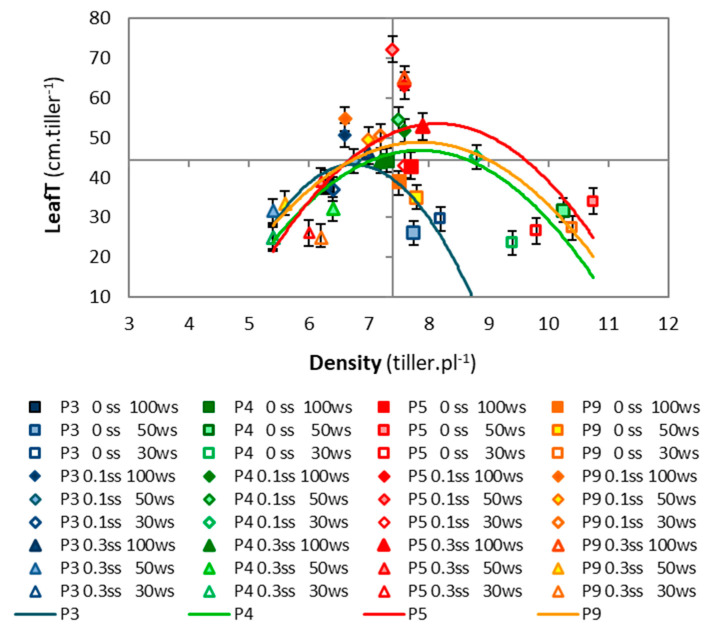
Relationship between Density and leaf length (LeafT) of four accessions (P3, P4, P5, P9) grown under three drought levels (0 WS, 50 WS, or 70 WS, corresponding to 100%, 50%, or 30% water capacity, respectively) combined with three salinity levels (0 SS, 0.1 SS, or 0.3 SS, corresponding to 0.0, 0.1, or 0.3 M NaCl, respectively). Data are means of five blocks (*n* = 5). Bars indicate the means standard error. Colored lines show the behavior of each accession. P3: y = −8.5x^2^ + 114.4x − 342.6, R^2^ = 0.76 *; P4: y= −3.8x^2^ + 59.9x − 188.3, R^2^ = 0.62 *; P5: y= −4.3x^2^ + 69.1x − 227.98, R^2^ = 0.46 ns; P9: y= −3.4x^2^ + 54.3x − 164.8, R^2^ = 0.39 ns (ns *p* > 0.0500, * *p* < 0.0500).

**Figure 4 plants-11-01548-f004:**
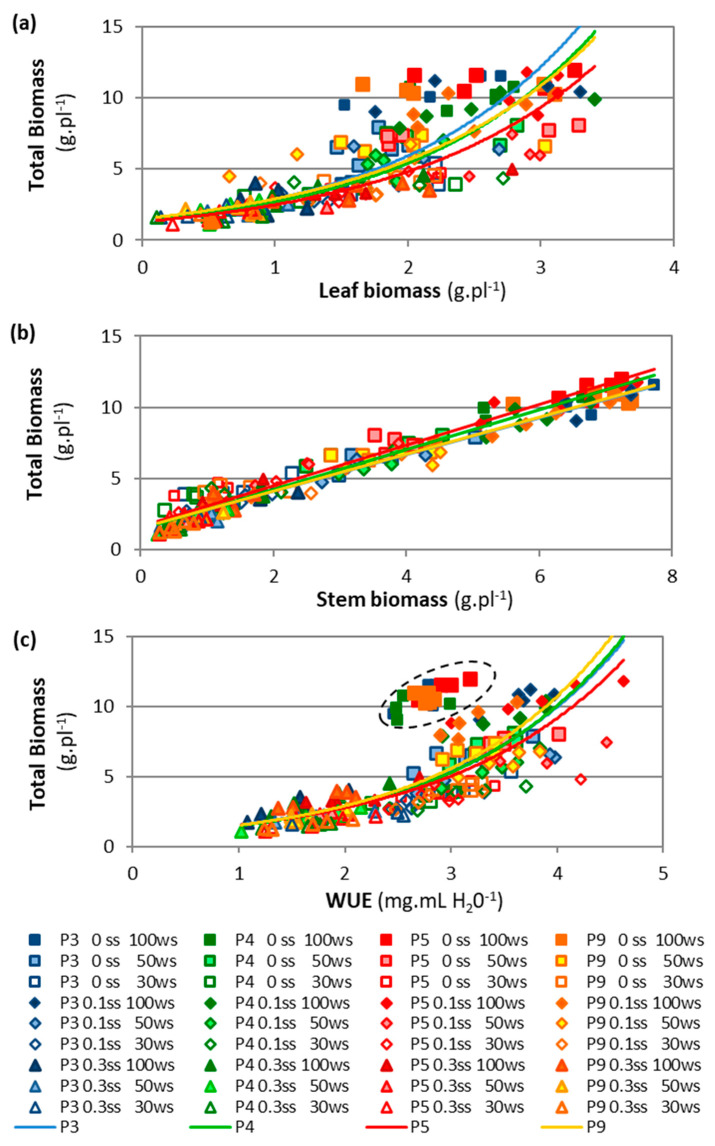
Relationship between total biomass and leaf biomass (**a**), stem biomass (**b**), or water-use efficiency (WUE) (**c**) of four accessions (P3, P4, P5, P9) grown under three drought levels (0 WS, 50 WS, or 70 WS, corresponding to 100%, 50%, or 30% water capacity, respectively) combined with three salinity levels (0 SS, 0.1 SS, or 0.3 SS, corresponding to 0.0, 0.1, or 0.3 M NaCl, respectively). Colored lines show the behavior of each accession. For leaf biomass, P3: y = 1.40e^0.72x^ R^2^ = 0.75 ***, P4: y = 1.32e^0.71x^ R^2^ = 0.79 ***, P5: y = 1.28e^0.66x^ R^2^ = 0.80 ***, P9: y = 1.49e^0.66x^ R^2^ = 0.60 ***; for stem biomass, P3: y = 1.29x + 1.50 R^2^ = 0.97 ***, P4: y = 1.40x + 1.44 R^2^ = 0.96 ***, P5: y = 1.43x + 1.63 R^2^ = 0.96 ***, P9: y = 1.30x + 1.51 R^2^ = 0.95 ***; and for WUE, P3: y = 0.83e^0.62x^ R^2^ = 0.60 ***, P4: y = 0.78e^0.64x^ R^2^ = 0.62 ***, P5: y = 0.84e^0.60x^ R^2^ = 0.62 ***, P9: y = 0.79e^0.65x^ R^2^ = 0.60 ***, (*** *p* < 0.001). The dotted parabola encloses the WUE behavior of each accession in the 0 and 0 SS treatments (d).

**Figure 5 plants-11-01548-f005:**
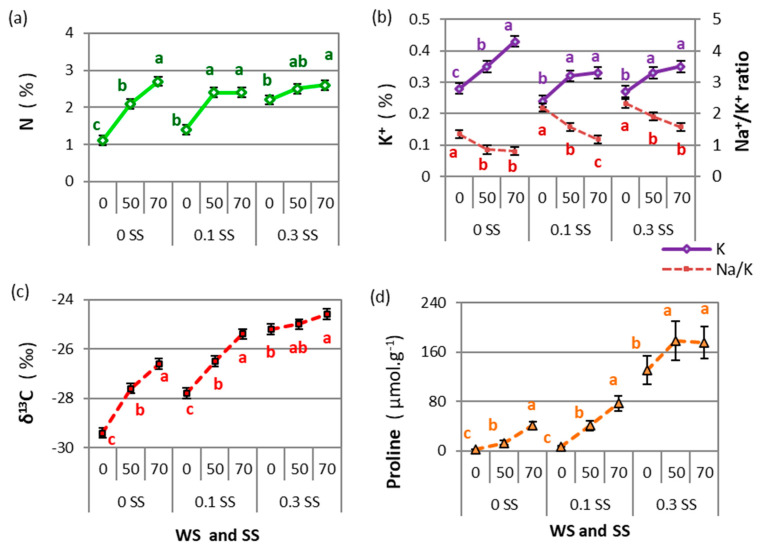
Drought and salinity interactions (WS and SS) for N concentration (**a**), K^+^ concentration and Na^+^/K^+^ ratio (**b**), stable isotopes of C (δ^13^C) (**c**), and free proline (**d**). For each variable, different letters show significant differences (LSD, *p* < 0.05) among drought levels (WS: 0, 50, or 70, corresponding to 100%, 50%, or 30% water capacity, respectively) for each salinity level (SS: 0, 0.1, or 0.3, corresponding to 0.0, 0.1, or 0.3 M NaCl, respectively). Points are means of 4 tall wheatgrass accessions and 5 blocks (*n* = 20). Bars indicate the means standard error.

**Figure 6 plants-11-01548-f006:**
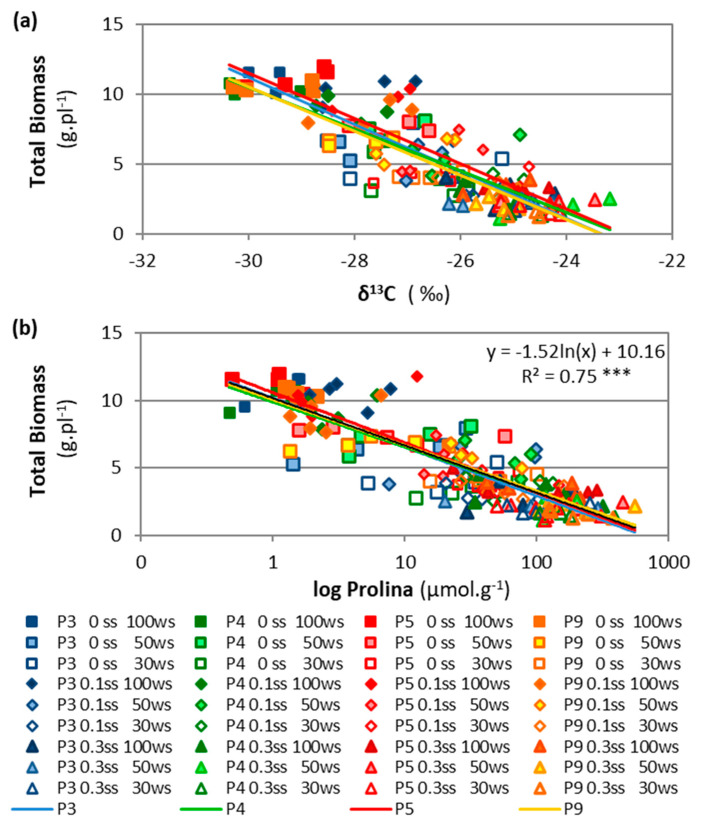
Relationship between total biomass and stable isotope of carbon (δ^13^C) (**a**) or free proline (**b**) for four accessions (P3, P4, P5, P9) grown under three drought levels (0 WS, 50 WS, or 70 WS, corresponding to 100%, 50%, or 30% water capacity, respectively) combined with three salinity levels (0 SS, 0.1 SS, or 0.3 SS, corresponding to 0.0, 0.1, or 0.3 M NaCl, respectively). Colored lines show the behavior of each accession. For δ^13^C, P3: y = −1.75x − 41.30 R^2^ = 0.69 ***, P4: y = −1.61x − 37.64 R^2^ = 0.76 ***, P5: y = −1.76x − 40.60 R^2^ = 0.70 ***, P9: y = −1.66x − 38.74 R^2^ = 0.78 ***; for proline, P3: y = 1.57ln(x) + 10.17 R^2^ = 0.63 ***, P4: y = 1.45ln(x) + 9.88 R^2^ = 0.77 ***, P5: y = 1.6157ln(x) + 10.57 R^2^ = 0.80 ***, P9: y = 1.47x + 10.06 R^2^ = 0.84 *** (*** *p* < 0.001). The black line shows the accession mean presented in Figure 6b.

**Figure 7 plants-11-01548-f007:**
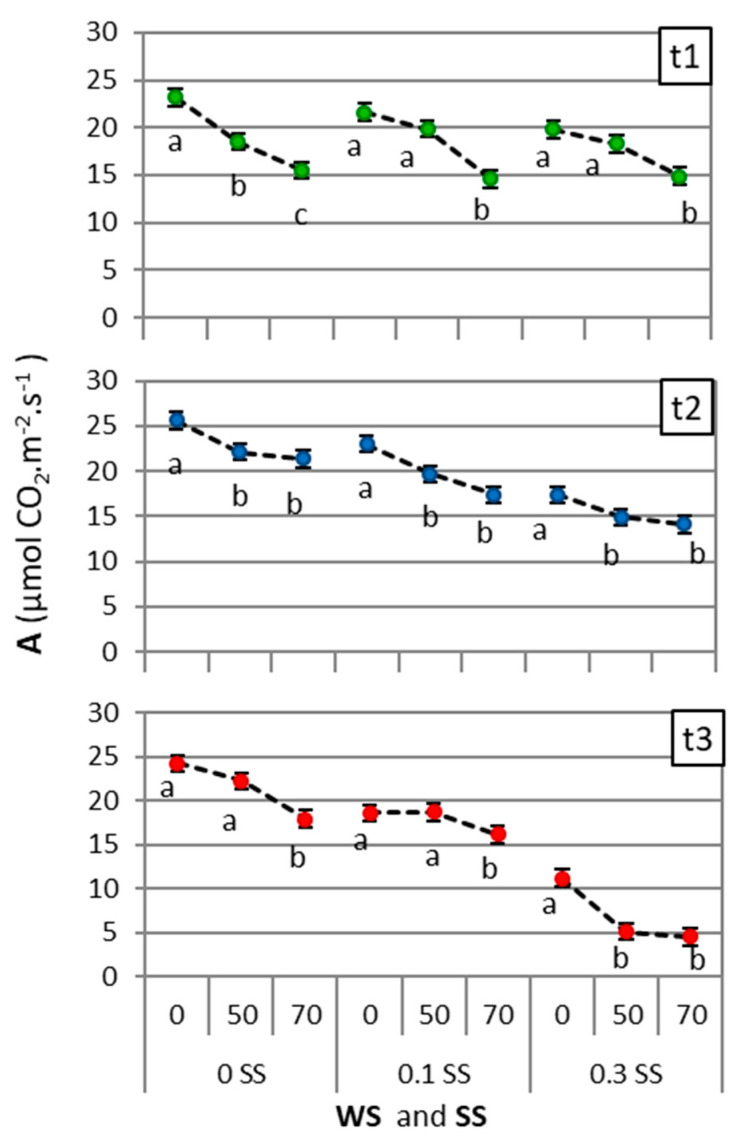
Effect of WS × SS × t interaction on net photosynthetic rate (A) for each time point (t1, t2, and t3, corresponding to 10, 45, and 85 days, respectively). Different letters indicate significant differences (*p* < 0.05) between drought levels (WS: 0, 50, or 70, corresponding to 100%, 50%, or 30% water capacity, respectively) for each salinity level (SS: 0, 0.1, or 0.3, corresponding to 0.0, 0.1, or 0.3 M NaCl, respectively). Points are means of 4 tall wheatgrass accessions and 4 blocks (*n* = 16). Bars indicate the means standard error.

**Figure 8 plants-11-01548-f008:**
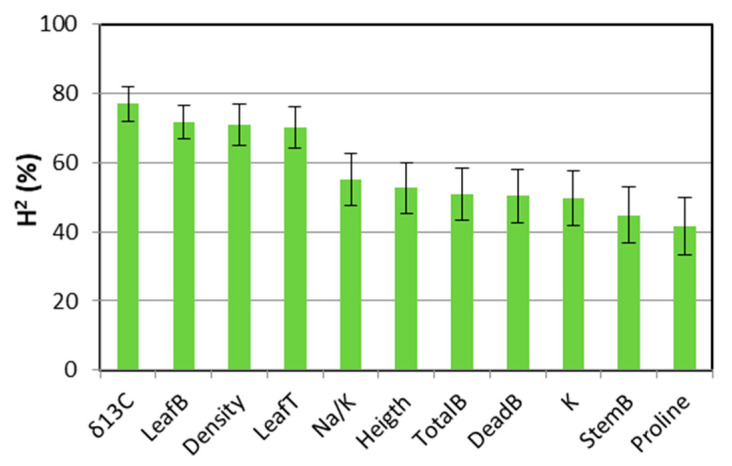
Broad-sense heritability (H^2^) and standard errors (bars) for morpho-agronomic, physiological, and isotopic variables: stable isotopes of C (δ^13^C); tiller density (Density); leaf length per tiller (LeafT); leaf (LeafB), stem (StemB), dead (DeadB), and total (TotalB) biomass dry weight; Na^+^/K^+^ ratio (Na/K); plant height (Height); and K^+^ concentration (K).

**Table 1 plants-11-01548-t001:** Tiller density, leaf length per tiller (LeafT), green leaf biomass (LeafB), green stem biomass (StemB), total biomass dry weight (TotalB), percentage of dead biomass (Dead%), plant height (Height), accumulated evapotranspiration (ETA), water-use efficiency (WUE), proportion of spiked tillers (Spike%), Na^+^/K^+^ ratio (Na^+^/K^+^), and stable isotopes of C (δ^13^C) of four tall wheatgrass accessions (P3, P4, P5, P9); Spike% values are given for the four accessions at each salinity level (0 SS, 0.1 SS, or 0.3 SS, corresponding to 0.0, 0.1, or 0.3 M NaCl, respectively).

	Density	LeafT	LeafB	Dead%	StemB	TotalB	Height	ETA	WUE		Spike%		Na^+^/K^+^	δ^13^C
**Accession**	**(tiller.pl^−1^)**	**(cm.tiller^−1^)**	**(g.pl^−1^)**	**(%)**	**(g.pl^−1^)**	**(g.pl^−1^)**	**(cm.pl^−1^)**	**(mLH_2_O pot^−1^)**	**mg.mL^−1^_H_2_O_**	**0 SS**	**0.1 SS**	**0.3 SS**		**(‰)**
P3	6.8 b	36.4 b	1.57 b	19.5 ab	2.93 a	5.27ab	66.9 a	1893 ab	2.66 b	53 a	42 b	8 a	1.44 b	−26.55 b
P4	7.7 a	39.2 ab	1.64 b	20.0 a	2.61 b	5.09 b	63.2 ab	1849 b	2.63 b	30 b	46 b	10 a	1.70 a	−26.48 ab
P5	7.9 a	44.1 a	1.91 a	18.7 ab	2.73 ab	5.53a	61.8 b	1861 b	2.83 a	40 b	34 c	8 a	1.40 b	−26.21 a
P9	7.3 ab	42.2 a	1.62 b	16.8 b	2.91 a	5.21b	64.4 ab	1904 a	2.62 b	40 b	55 a	9 a	1.58 ab	−26.60 b
ee	0.3	2.5	0.07	1.4	0.12	0.13	1.6	42	0.06	4.1	4.0	3.6	0.09	0.16

Data are means of 3 drought levels, 3 salinity levels, and 5 blocks (*n* = 45), except for Spike%, which is means of 3 drought levels and 5 blocks (*n* = 15). Different letters in the same column indicate significant differences (LSD, *p* < 0.05). ee, mean standard error; pl, plant.

**Table 2 plants-11-01548-t002:** Tall wheatgrass accessions. Collection data from Active Germplasm Bank of Estación Experimental Agropecuaria Balcarce of the INTA, Argentina (BAL).

Accession	P3	P4	P5	P9
Latitude, longitude	38°30′ S, 58°45′ W	39°24′ S, 65°36′ W	39°24′ S, 65°36′ W	38°44′ S, 62°33′ W
Nearest town, province	Necochea, Buenos Aires	Lamarque, Río Negro	Lamarque, Río Negro	Bahía Blanca, Buenos Aires
Köppen ^§^	Cfb	BSk	BSk	Cfa
Climate	Temperate oceanic	Semiarid	Semiarid	Temperate transitional
Precipitation	840 mm year^−1^	266 mm year^−1^	266 mm year^−1^	583 mm year^−1^
Great Group Soil	*Argiudolls.*	*Torrifluvents.*	*Torrifluvents*.	*Haplustolls.*
Soil type, pH ^#^	Non-saline/non-alkaline soil, pH ≅ 7.0	Saline–alkaline soil, pH: 9.0	Non-saline/non-alkaline soil, pH: 7.5	Saline alkaline soil, pH: 9.5
Environment	Roadside grassland	Grassland, with Distichlis spicata	Roadside grassland of irrigated fields	Natural grasslands, with Distichlis spicata
Collection BAL ^†^	Nu + Alo 338	CIB 118	CIB 117	CIB 114

^§^ Köppen climate classification. ^#^ Soil PH content corresponding to the 0–100 mm depth layer. ^†^ Collector’s code (Nu, Alo, CIB) and entry number.

## Data Availability

Data are contained within the article or Appendix A. Raw data are available by contacting the corresponding author.

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
