# Peer review of "Ecophysiological Responses of Tall Wheatgrass Germplasm to Drought and Salinity"

_plants, 2022, doi:10.3390/plants11121548_

Round 1
Reviewer 1 Report
I have an impression that the manuscript has been thoroughly improved and corrected.
I still found abbreviations used for different salinity treatments confusing. If these indeed are decimals, zero needs to be included. In any case, no term should start with a period (".").
Author Response
Thank you for the valuable comments and suggestions. They were contemplated and incorporated in the manuscript. The abbreviations used for different salinity treatments were corrected in texts, tables and figures. Considering your comments, this manuscript was markedly improved.
Authors.
Reviewer 2 Report
Borrajo et al. provided a revised version of their manuscript on the morpho-agronomic, physiological and isotopic response of a C3 forage grass, Thinopyrum ponticum, from four germplasm accessions, to different water and salt stress combinations. Most of my comments and concerns to the original version were addressed, and I feel the manuscript to be much clearer.
I have only a few suggestions on the new text:
Line 18: Add an endpoint after “greenhouse”
Line 19: “moderate stress” needs to be more precise.
Line 55-56: Change “CO2” to “CO2”
Line 59: Remove the extra space between “19 and 21”
Line 75: Remove the comma after “33”
Lines 14, 111, and 125: The same sentence was repeated in such lines. Please reformulated in order to reduce repetition.
Lines 132-134: The sentence was not totally true (there were no differences between drought treatments”. Please revise it.
Lines 134-135: This last sentence is not a result. Please move it to the Discussion section or remove it.
Lines 137-138: This sentence does not make sense here. Please move it to the next section.
Line 144: Remove the comma after “(SS)”.
Line 156: “moderate stress” of drought or salt or both?
Line 179: (DeadB) should be placed after “Dead biomass”.
Line 293-294: This sentence does not make sense here. Please move it to the next section.
Line 410: Changed “to” to “under”.
Line 551 and 559: Avoid repetition of “meanwhile”.
Author Response
Response to Reviewer 2
Line 18: Add an endpoint after “greenhouse”.
It was corrected.
Line 19: “moderate stress” needs to be more precise.
It was modified, lines 17-18.
Line 55-56: Change “CO2” to “CO2”.
They were modified, lines 53 and 54.
Line 59: Remove the extra space between “19 and 21”.
It was removed.
Line 75: Remove the comma after “33”.
It was removed.
Lines 14, 111, and 125: The same sentence was repeated in such lines. Please reformulated in order to reduce repetition.
The sentences were modified to reduce repetition in lines 11-14 and 122-127.
Lines 132-134: The sentence was not totally true (there were no differences between drought treatments”. Please revise it.
The sentence was corrected, lines 131-132.
Lines 134-135: This last sentence is not a result. Please move it to the Discussion section or remove it.
The sentence was modified, lines 132-134.
Lines 137-138: This sentence does not make sense here. Please move it to the next section.
This sentence was moved to the next section, lines 142-143.
Line 144: Remove the comma after “(SS)”.
It was removed.
Line 156: “moderate stress” of drought or salt or both?
This sentence was modified, lines 154-156.
Line 179: (DeadB) should be placed after “Dead biomass”.
It was moved, line 180.
Line 293-294: This sentence does not make sense here. Please move it to the next section.
This sentence was moved to the next section, lines 304-305.
Line 410: Changed “to” to “under”.
It was changed, line 412.
Line 551 and 559: Avoid repetition of “meanwhile”.
It was modified, line 561.
Thank you for the valuable comments and suggestions, considering your comments, this manuscript was markedly improved.
Authors.